# Effect of Titanium Modification on Microstructure and Impact Toughness of High-Boron Multi-Component Alloy

**Xiangyi Ren** [1,*]**, Shuli Tang** [2]**, Hanguang Fu** [3] **and Jiandong Xing** [4]

1   State Key Laboratory of Performance and Structural Safety for Petroleum Tubular Goods and Equipment Materials, CNPC Tubular Goods Research Institute, Xi'an 710077, China
2   Xi'an Microelectronic Technology Institute, Xi'an 710065, China; shulitangxjtu@126.com
3   Research Institute of Advanced Materials Processing Technology, School of Materials Science and Engineering, Beijing University of Technology, Beijing 100124, China; hgfu@bjut.edu.cn
4   State Key Laboratory for Mechanical Behavior of Materials, School of Materials Science and Engineering, Xi'an Jiaotong University, Xi'an 710049, China; jdxing@139.com
*   Correspondence: mmerenxiangyi@126.com; Tel.: +86-139-1996-4450

**Abstract:** This work investigated the microstructure and mechanical property of high-boron multi-component alloy with Fe, B, C, Cr, Mo, Al, Si, V, Mn and different contents of Ti. The results indicate that the as-cast metallurgical microstructure of high-boron multi-component alloys consist of ferrite, pearlite and borocarbide. In an un-modified alloy, continuous reticular structure of borocarbide is observed. After titanium addition, the structure of borocarbide changes into a fine and isolated morphology. TiC is the existence form of titanium in the alloy, which acts as the heterogeneous nuclei for eutectic borocarbide. Moreover, impact toughness of the alloy is remarkably improved by titanium modification.

**Keywords:** high-boron multi-component alloy; borocarbide; titanium; misfit degree; heterogeneous nuclei





## 1. Introduction

In the steel rolling industry, the roll is an important component. Considerable consumption of rolls wastes resources and energy severely. For solving this problem, it is necessary to investigate roll materials with remarkable wear resistance. Previously, numerous metal materials with high hardness were successfully applied in the roll-making industry [1–4]. For a long time, high-chrome cast iron acts as the widest used wear-resistant roll material [5–8]. Factories in Europe started to use cast iron roll in the early 1990s. Subsequently, high-speed steel rolls became widely used in the year 1994 [2]. Carbide in high-speed steel possesses excellent hardness, hardenability, toughness and wear resistance. Thus, high-speed steel was the best material for roll manufacturing at that time. Soon afterwards, high-vanadium high-speed steel, which contains more carbon and vanadium, was invented in which numerous isolated MC (M refers to metal element) carbide particles were found, bringing about considerable augmentation of thermal and mechanical property [9–14]. However, disadvantages are exposed with its wide utilization, such as high cost and poor toughness and thermal fatigue resistance [15]. Thus, it is important for the investigation of new-type roll material with better properties.

China is a country with rich boron resources. In these years, numerous research into boron-added alloys were carried out, which reveals that boron-added cast iron presents good mechanical properties due to the hard and thermal-stable boride in the microstructure [16–23]. Furthermore, addition of boron instead of expensive alloying elements reduces the production cost and simplifies the manufacturing process [24–30]. High-boron multi-component alloy is a new-type wear-resistant material. Hardness of borocarbide in this alloy is 1700 HV approximately, which is lower than that of vanadium carbide (VC, about 2000 HV) [21,31]. It is promising that high-boron multi-component alloys can be used

widely as roll material. However, in high-boron multi-component alloys, continuously distributed borocarbide separates the matrix severely [29,32], making it unable to support the borocarbide to resist the abrasion by means of its toughness. Some of the researchers investigated whether the structure of heat-treated borocarbide is isolated, but the results were unsatisfying [33,34].

Modification treatment is a widely applied technology to refine the carbide in alloys [35–41]. According to the previous works, microstructure of high-speed steel can be changed by some kind of particular element so that the property of which can also be improved. In previous investigations [42–51], high speed steel used surface-active elements (sodium, potassium and aluminum) and/or heterogeneous nuclei elements (cerium, yttrium and titanium) as modifying elements. It was revealed that these elements effectively refine and/or spheroidize the carbide. For isolating the borocarbide in high-boron multi-component alloy, modifying elements are also used on refining or spheroidizing borocarbide in some researches. As a kind of effective modifying element as well as alloying element, titanium is widely used by investigators to improve the microstructure and property of cast iron and boron-added ferrous alloys. Chung et al. indicated that in high chromium cast iron with titanium added as an alloying element, wear resistance of which is improved because of the refinement of eutectic carbides [52]. He et al. [17], Liu et al [36]. and Shi et al [40] suggested that boride in an Fe-B alloy can be refined and sphereoidized by titanium modification. As a promising roll material, high-boron multi-component alloy should possess good property through ameliorating the microstructure. This research used titanium as a modifying element and studied the effect of titanium on solidification structure and the mechanical properties of high-boron multi-component alloys with the composition of Fe-B-C-Cr-Mo-Al-Si-V-Mn-Ti.

## 2. Experimental Process

### 2.1. Experimental Materials

All of the materials for high-boron multi-component alloy casting were smelted to approximately 1600 °C in air in 8 kg induction furnace. First of all, a ladle in which there was liquid steel and FeTi70 Modifier particles in the diameter of 5 mm was filled. Two minutes later, liquid steel with a soluted modifier was discharged into a pre-heated mold under 1500–1550 °C. Figure 1 presents the shape and size of the mold. The metallurgical analysis used 10mm cube specimens and 10 mm × 10 mm × 55 mm impact specimens (without notch). All of the specimens were cut from the center of the ingot, where the estimated cooling rate was 40 °C/s. Table 1 shows the addition amount of modifier in each specimen. The heat treatment process was 1050 °C, 2 h, quenching + 500 °C, 1 h tempering.

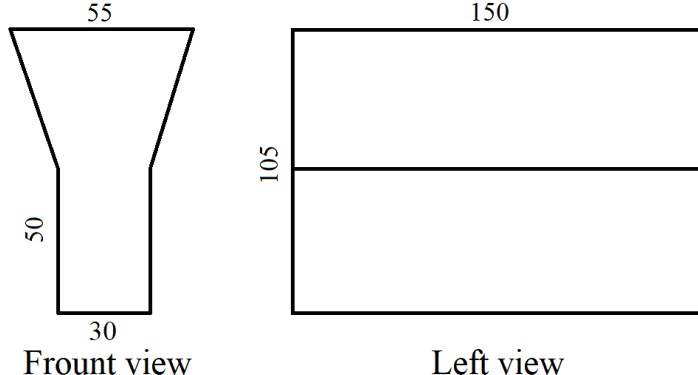

**Figure 1.** Shape and size of mold cavity (mm).

**Table 1.** Alloy symbols and the corresponding modifier contents of four specimens (wt.%).

| Alloy Symbol | M | T1 | T2 | T3 |
|:---:|:---:|:---:|:---:|:---:|
| Modifier content | 0.0 | 0.3 | 0.6 | 0.9 |

### 2.2. Analysis and Characterization

X-ray flourescence (XRF) and energy dispersion spectrum (EDS) analysis were used to determine the chemical composition of the studied alloys and FeTi70 modifier. The results tested by XRF are shown in Tables 2 and 3. Microstructure of metallurgical specimens was analyzed by optical microscope (OM), X-ray diffraction (XRD), scanning electron microscopy (SEM, Tescan VEGA II XMU, TESCAN, Brno, Czech Republic) and electron probe microanalyzer (EPMA, JXA-8100 JEOL Ltd., Tokyo, Japan). The characterization of crystal structure and relationship of lattice orientation of phases was analyzed by transmission electron microscopy (TEM, JEM-2100, JEOL USA Inc., Peabody, Massachusetts) and selected area (electron) diffraction (SAD). Hardness of specimens was measured by Rockwell-hardness tester (Great Chang'an Test Co. Ltd., Dongguan, China). Microhardness of different microstructures in each specimen can be measured by Vickers-hardness tester (Hansheng Precision machinery Co. Ltd., Dongguan, China). Metallographic samples were polished by sandpaper and etched by 5% $HNO_3$ solution. For carrying out the quantitative analysis, specimens were colored by Carlin corrosive. Metallurgical surfaces of alloy T2 and M were deeply etched by saturated $FeCl_3$ solution for XRD analysis. The JB30A impact testing device (Hansheng Precision machinery Co. Ltd., Dongguan, China) was used to measure the impact toughness tests of heat-treated specimens.

**Table 2.** Chemical compositions of alloys (wt.%).

| Specimens | B | C | Cr | Mo | Al | Si | V | Mn | Ti | Fe |
|:---:|:---:|:---:|:---:|:---:|:---:|:---:|:---:|:---:|:---:|:---:|
| M | 1.95 | 0.38 | 5.87 | 3.86 | 0.85 | 0.88 | 0.85 | 0.46 | 0.00 | Bal. |
| T1 | 1.97 | 0.34 | 5.29 | 3.79 | 0.80 | 0.89 | 0.92 | 0.51 | 0.21 | Bal. |
| T2 | 1.96 | 0.36 | 5.12 | 3.85 | 0.86 | 0.96 | 0.91 | 0.52 | 0.41 | Bal. |
| T3 | 1.93 | 0.33 | 5.70 | 3.78 | 0.82 | 0.81 | 0.85 | 0.45 | 0.54 | Bal. |

**Table 3.** FeTi70 modifier composition (wt.%).

| Si | Ti | Al | Mn | C | Fe |
|:---:|:---:|:---:|:---:|:---:|:---:|
| 2.0 | 68.0 | 4.5 | 2.5 | 0.2 | Bal. |

The quantitative analysis includes volume fraction $V_V$, shape factor $K$ and Feret diameter $dF$ [53,54]. Software ImageJ (1.49v, National Institutes of Health, Bethesda, MD, USA) can be used to realize the quantitative analysis. The Stereological Equiation (1) suggests that the volume fraction can be substituted by area fraction $A_A$.

$$V_V = A_A \tag{1}$$

The calculation of the shape factor $K$ is shown in Equation (2), in which $A$ refers to the area ($\mu m^2$) and $L$ refers to the perimeter ($\mu m$). $K \in (0, 1)$. The larger $K$ is, the closer to the circle tested the phase is.

$$K = \frac{4\pi A}{L^2} \tag{2}$$

The Feret diameter $dF$ is the average width of a phase at not less than 40 different directions. Figure 2 presents its calculating process.

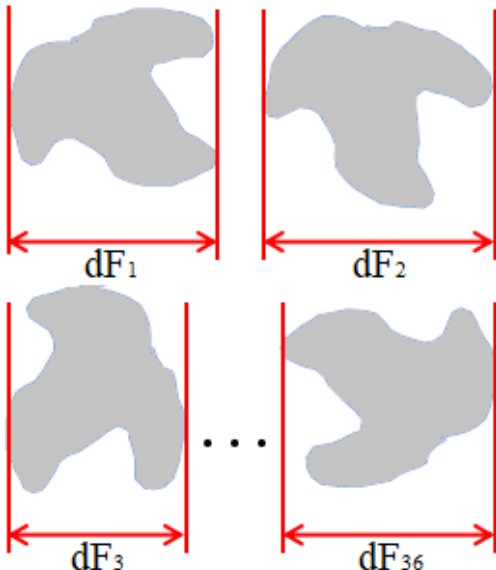

**Figure 2.** Feret diameter calculating process.

## 3. Results and Discussion

### 3.1. Effect of Titanium Concentration on Microstructure of High-Boron Multi-Component Alloy

Figure 3 shows the as-cast microstructure of a high-boron multi-component alloy, which is unmodified and modified with different titanium concentrations. It is observed that the matrix in an alloy consists of ferrite (light area) and pearlite (dark area). In Figure 3a, borocarbide (reticular area) in alloy M presents a relatively large size. With titanium addition, borocarbide is obviously refined, as presented in Figure 3b–d. Some of the previous works [36,40,42,44,45] have indicated that titanium in liquid steel forms plenty of intermetallic compound particles with other elements. These particles act as heterogeneous nuclei, providing a base for nucleation of borocarbide and significantly raising its nucleation rate. When the amount of borocarbide is constant, the higher nucleation rate that borocarbide has, the more isolated borocarbide will appear. Hence, the size of borocarbide is decreased. For certifying the formation of titanium-included compounds in modified alloys and eliminating the interference of matrix, the deep-etched alloy M and T2 are analyzed by XRD. Their results are shown in Figure 4. Figure 4a presents the morphology of the deep-etched alloy M, indicating that the matrix is completely erased. In Figure 4b, three kinds of phases, $\alpha$-Fe and borocarbide $M_2(B,C)$ and $M_3(B,C)$ (M refers to metal elements in alloy except Al) are detected in both alloy M and T2. In the curve of alloy T2, obvious diffraction peaks of phase TiC can be observed. Formation of TiC proves that one of the conditions for titanium to take effect as a modifying element is required.

Figure 5 shows the heat-treated morphologies of the alloys. Excellent hardenability is presented in the boron-added matrix [32]. Consequently, the matrix completely transforms to a martensite structure. In unmodified and modified alloys, angles of borocarbide are blunted and continuous borocarbide is disconnected. The size of borocarbide in modified alloys is smaller, and the borocarbide-matrix interface is more curved and less stable, so there are more locations that can be disconnected. In alloy M, borocarbide still keeps the continuous network, the shape of which is quite similar to the as-cast structure. Boron possesses higher solubility in austenite than in ferrite [38], and during martensite transformation, boron atoms near the interface diffuse into the matrix. Thus, the sharp angle of borocarbide is blunted, and the narrow place of borocarbide is broken.

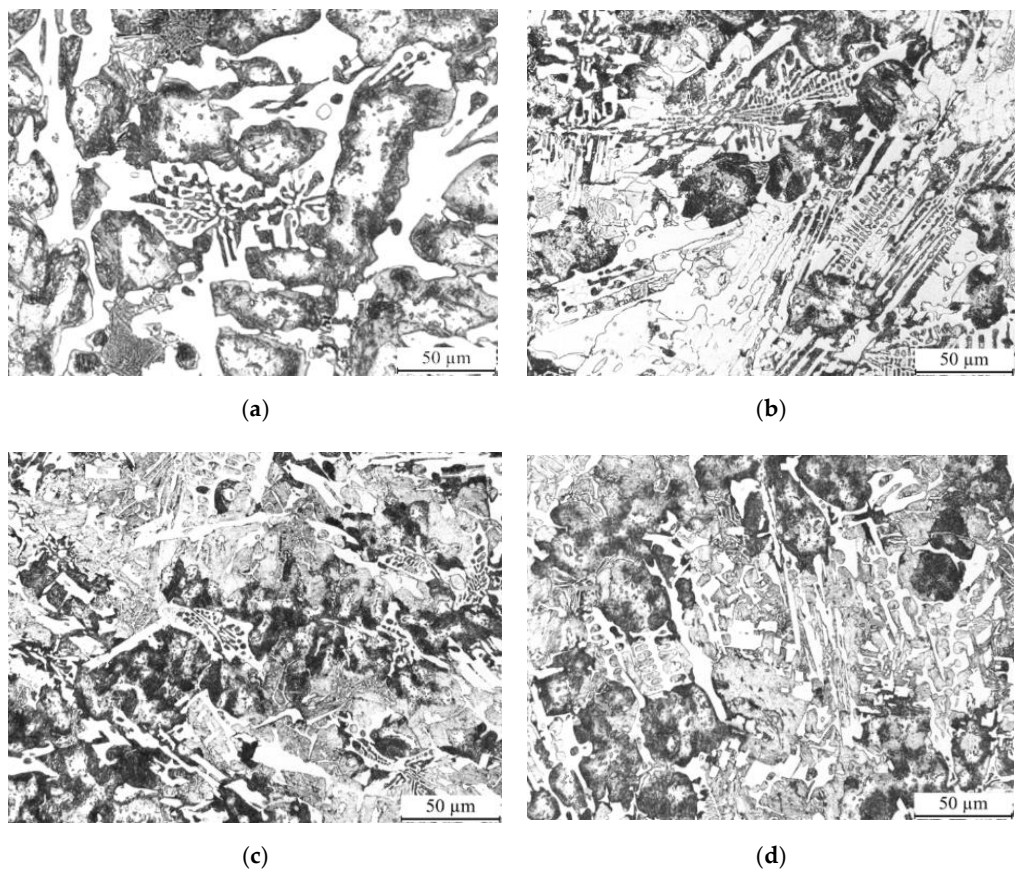

**Figure 3.** As-cast morphology of high-boron multi-component alloy with different Ti contents: (**a**) M, 0.0 wt.%, (**b**) T1, 0.2 wt.%, (**c**) T2, 0.4 wt.%, (**d**) T3, 0.6 wt.%.

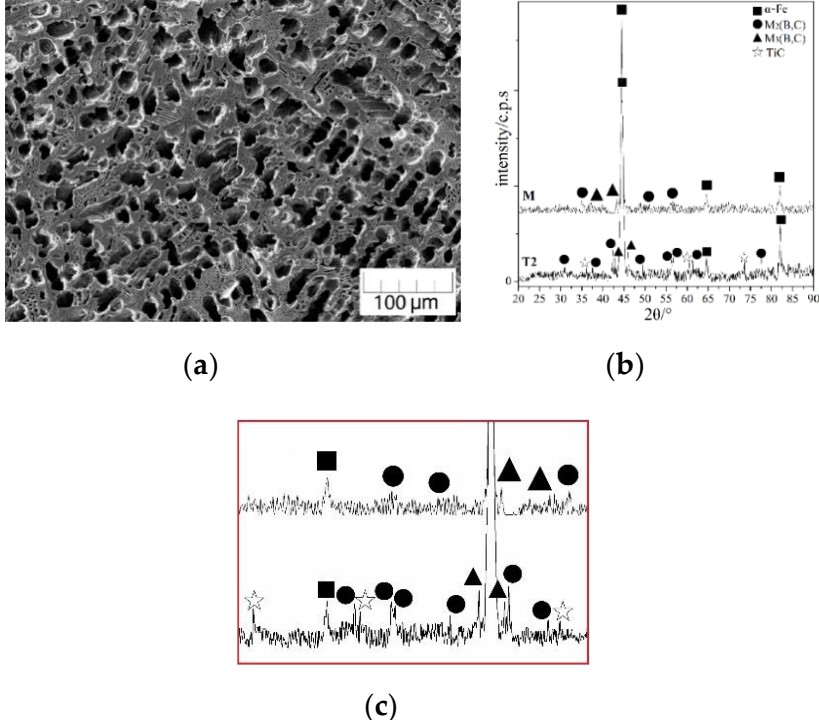

**Figure 4.** Morphology deep-etched as-cast alloy M (**a**), X-ray diffraction (XRD) patterns of deep-etched alloy T2 and M (**b**) and part details of XRD patterns (**c**).

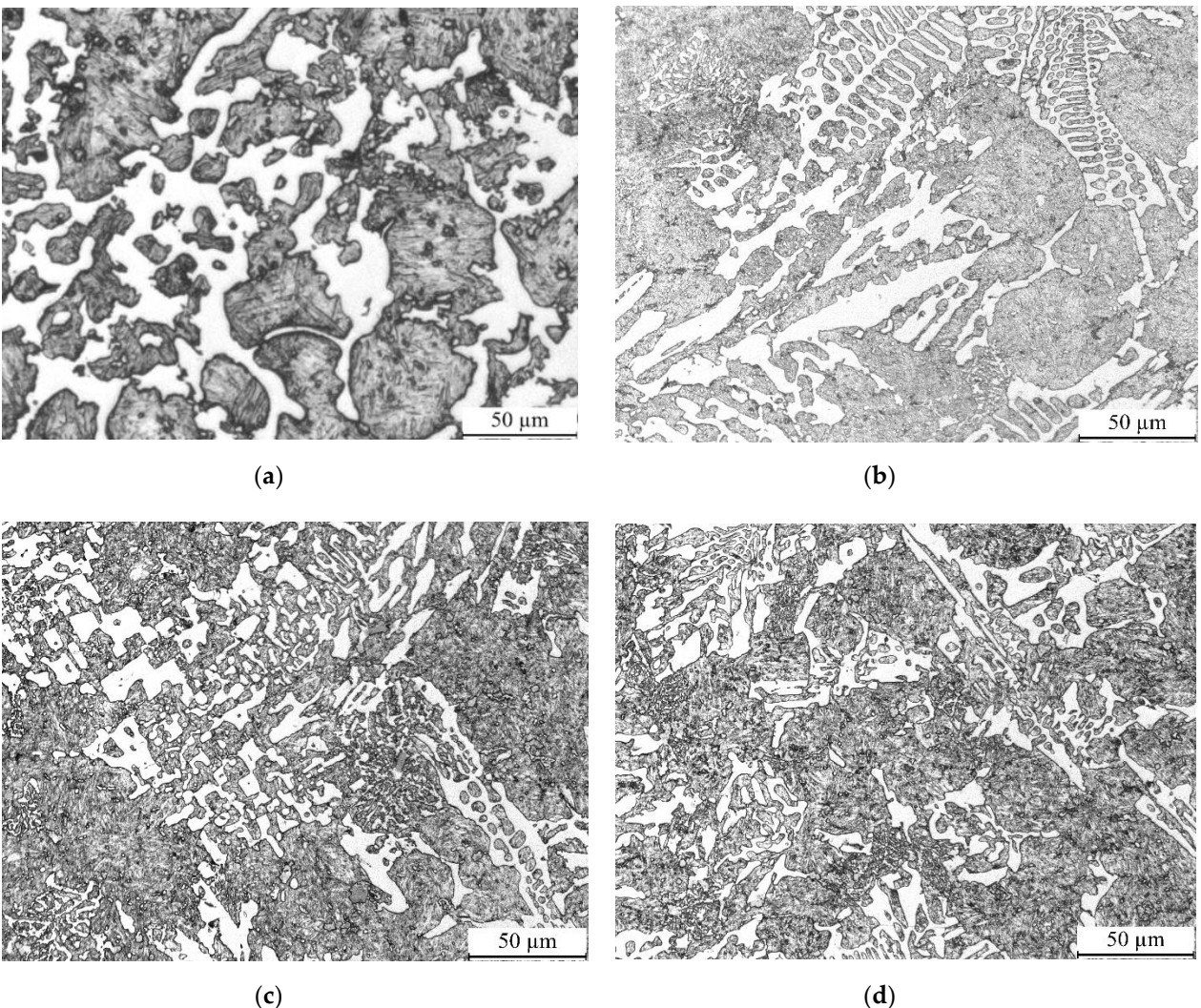

**Figure 5.** Morphologies of heat-treated high-boron multi-component alloy with various Ti contents: (**a**) M, 0.0 wt.%, (**b**) T1, 0.2 wt.%, (**c**) T2, 0.4 wt.%, (**d**) T3, 0.6 wt.%.

The modifying effect of titanium on borocarbide morphology improvments cannot be accurately characterized by normal observation. Thus, it is necessary to use the quantitative analysis to reveal the varying law of borocarbide. The results of the quantitative analysis are presented in Figure 6. In Figure 6a, borocarbide volume fraction diminishes with the increase of titanium content in the as-cast alloys. From reference [55], after slight titanium addition in the ferro-boron alloy, the amount of borocarbide descends when titanium content increases. Besides, borocarbide is coarsened after tempering treatment, and thus the volume fraction increases [56]. Borocarbide is significantly spheroidized and refined after titanium modification, which can be seen in Figure 6b,c. Moreover, the heat-treated borocarbide is further spheroidized and refined.

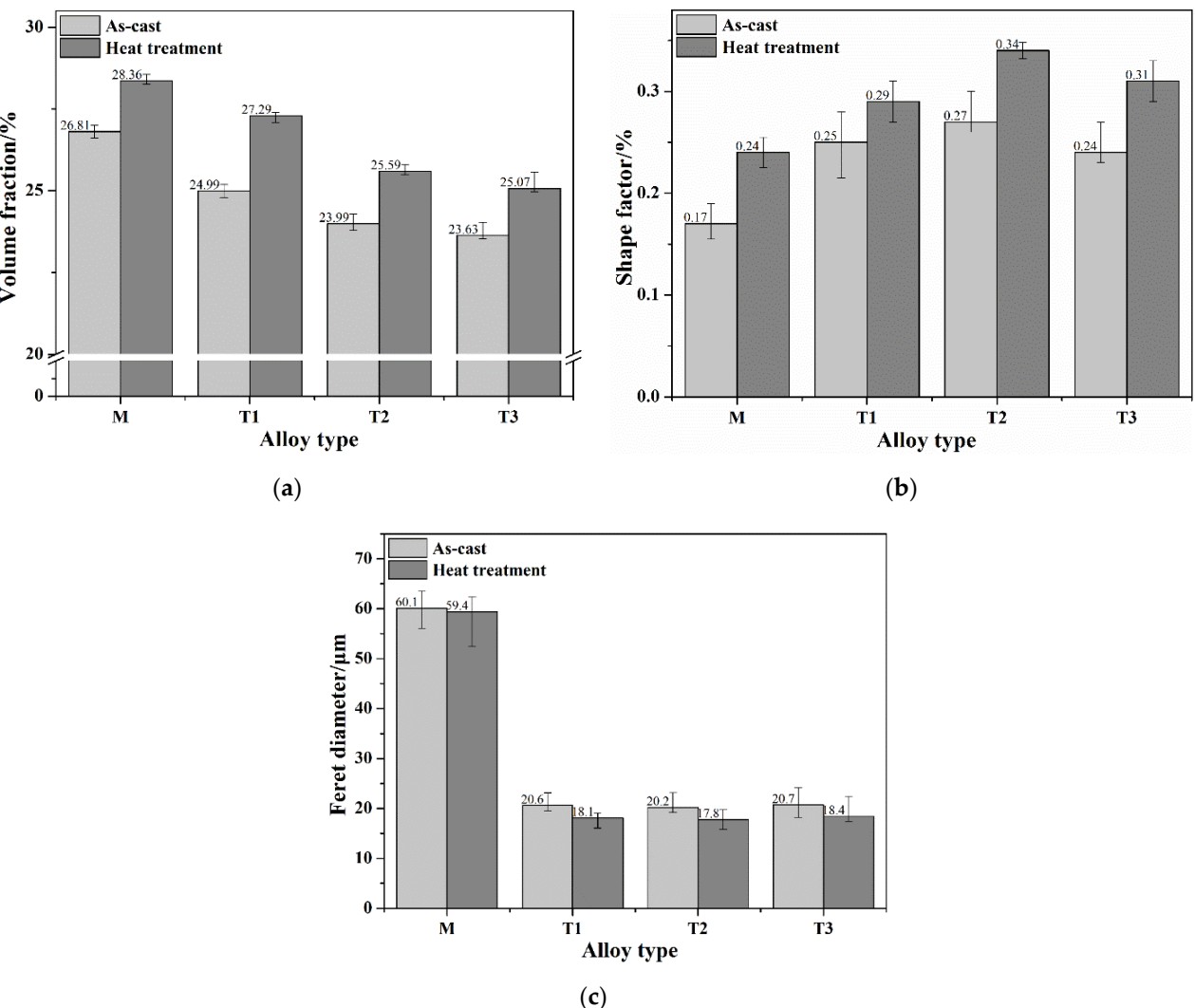

**Figure 6.** Quantitative calculating results of borocarbide in high-boron multi-component alloy with various Ti contents: (**a**) $V_v$, (**b**) $K$, (**c**) $dF$.

### 3.2. Analysis of Titanium-Modifying Mechanisms in High-Boron Multi-Component Alloy

According to the results above, titanium distributes in the alloy in the form of TiC compounds. Three essential conditions are used to judge if TiC is an effective heterogeneous nuclei: precipitating priority, low misfit degree and dispersed distribution [47]. The thermodynamic estimation of TiC precipitating order is carried out. From the thermodynamic data of reference [57], the formation of TiC in liquid steel is as follows:

$$C\ (s) = [C]\ \Delta_r G_1 = 22{,}590 - 42.26\ T\ kJ/mol \tag{3}$$

$$Ti\ (s) = [Ti]\ \Delta_r G_2 = -25{,}100 - 44.98\ T\ kJ/mol \tag{4}$$

$$Ti\ (s) + C\ (s) = TiC\ (s)\ \Delta_r G_3 = -186{,}600 + 13.2\ T\ kJ/mol \tag{5}$$

In liquid steel, forming the reaction of TiC can be obtained by Equation (5)–Equation (4)–Equation (3):

$$[Ti] + [C] = TiC\ (s)\ \Delta_r G_4 = -182{,}290 + 99.79\ T\ kJ/mol \tag{6}$$

Reference [32] indicates that the melting point of high-boron multi-component alloy is 1602 K. When *T* is 1602, it is calculated that value of $\Delta_r G_4$ is −224.26 kJ/mol. Hence, it is inferred that the formation of TiC under this temperature is spontaneous, which is practica-

ble for prior precipitation. Moreover, reference [47] suggests that under 1602 K, content of TiC is higher than its solubility in the studied alloy. In conclusion, the prior formation of TiC is possible with the precipitation of Ti from liquid alloy before solidification.

For exploring the distribution of TiC in the alloy, the SEM BSE (back scattered electron) morphology of alloy T2 is observed and is shown in Figure 7. In Figure 7a, it is obviously seen that TiC distributes in the form of isolated particles dispersedly. Besides, there are no isolated TiC particles in the middle of the matrix. Some of the particles are inside the borocarbide (Figure 7b), others are at the interface of the matrix and borocarbide (Figure 7c). Figure 8a is the location of EPMA point scanning for analyzing the composition of TiC particles, the results of which are shown in Table 4. It is observed that the atomic ratio of Ti and C is 1:1 approximately. Moreover, some other metallic elements are found such as V, Mo, Cr, Mn and Fe because of the solid solution. Figure 8b presented the EPMA linear scanning results through the TiC particle. By observing the distributing curves of each metallic element, content of V and Mo in TiC is higher than that in borocarbide. Besides, Mo mainly distributes near the interface of TiC and borocarbide, which leads to the different brightness of TiC under SEM BSE observation. This is because the diffusion is more difficult for Mo atoms which have a larger atomic radius.

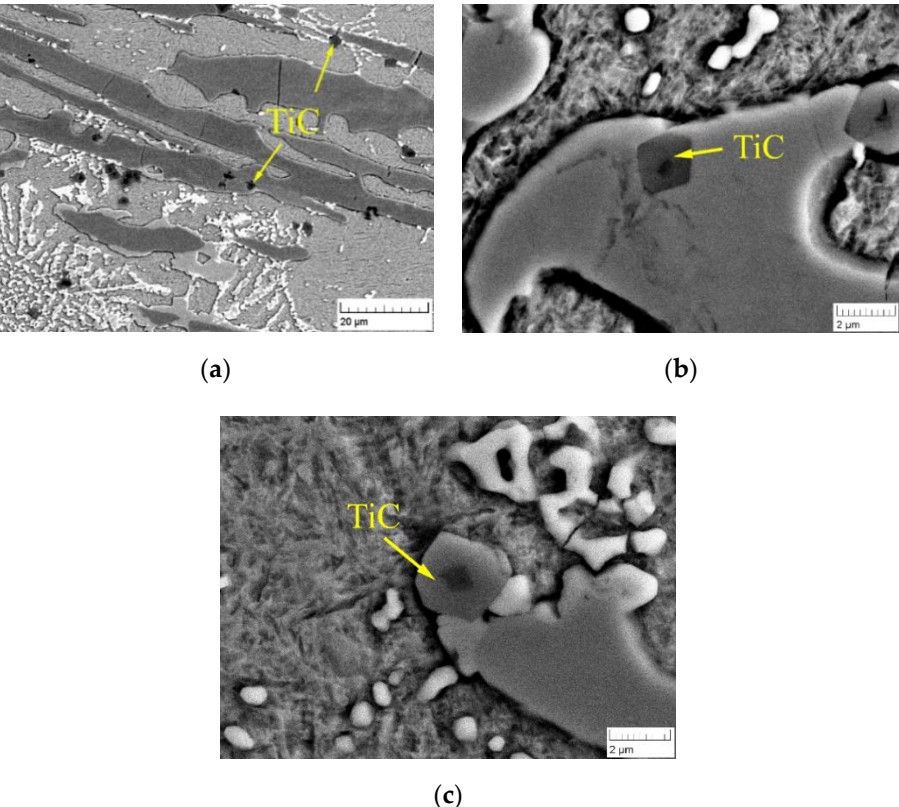

(**a**)　　　　　　　　　　　　　　　　　　　(**b**)

(**c**)

**Figure 7.** Distribution of TiC particles in heat-treated alloy T2 (**a**), TiC particle inside the borocarbide (**b**) and particle at the interface of matrix and borocarbide (**c**).

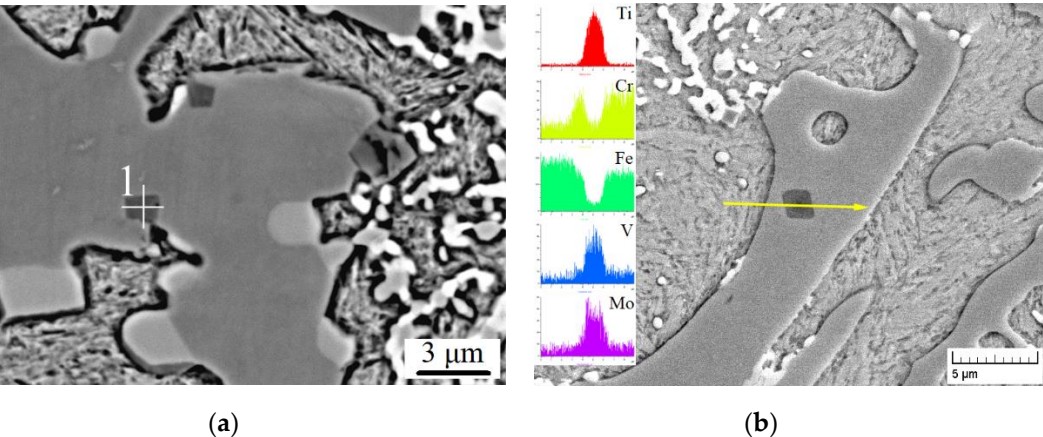

**Figure 8.** Distribution of alloying elements in TiC particle: (**a**) Point composition analysis, (**b**) Linear scanning results.

**Table 4.** EPMA results of point 1 (at.%).

| Point Number | C | Cr | Mo | V | Mn | Ti | Fe |
|---|---|---|---|---|---|---|---|
| Point 1 | 54.95 | 0.27 | 0.92 | 2.79 | 0.05 | 40.13 | 0.89 |

The misfit degree is used to determine whether one kind of phase is the effective heterogeneous nuclei for another phase. Figure 9 shows the lattice structures of TiC, austenite and boride $Fe_2B$. It is seen that structure of TiC, austenite is a face-centered cube (fcc) and $Fe_2B$ is a body-centered tetragonal structure, the lattice parameter of which is presented in Table 5. The misfit degree $\delta$ is calculated by the formula as follows [54]:

$$\delta_{(hkl)n}^{hkls} = \sum_{i=1}^{3} \frac{\left( \left| d[uvw]_s^i \cos\theta - d[uvw]_n^i \right| \right) / d[uvw]_n^i}{3} \times 100\% \tag{7}$$

Here, the $\delta$-misfit degree of two phases (*hkl*) is the crystallographic plane index, [*uvw*] is the crystal orientation index. *d* is the atomic length along [*uvw*]. *s* and *n* are the symbols of two different phases. $\theta$ is the included angle of two crystal orientations. The critical value of $\delta$ is under 12%. After the calculation of various lattice combinations of TiC with austenite and $Fe_2B$, it is found that the lowest misfit degree of TiC and austenite is 12.5%, which is from $(100)_{TiC} / / (110)_\gamma$. The lowest misfit degree of $M_2(B,C)$ and TiC is 9.3%, which is from $(110)_{TiC} / / (110)_{M2(B,C)}$. From these results, it can be proved that TiC particle is the ineffective nuclei for primary austenite and effective for $M_2(B,C)$.

The TEM and SAD analysis is used to verify the concordance of theoretical calculation and actual conditions, the results of which are shown in Figure 10. SAD patterns of region A and C reflect the lattice structure of TiC and $M_2(B,C)$, data in Table 5 are details of them. Region C is the combined pattern of TiC and $M_2(B,C)$. Normally, in a combined SAD pattern, when corresponding spots of two lattice and centric spots are detected to be alined, it can be proved that two crystallographic planes have an orientation relationship. Spots $(110)_{TiC}$, $(110)_{M2(B,C)}$ and the centric spot are detected to be alined obviously. This proves the orientation relationship $(110)_\gamma / / (110)_{M2(B,C)}$. These experimental results correspond to the results that are shown in Table 6.

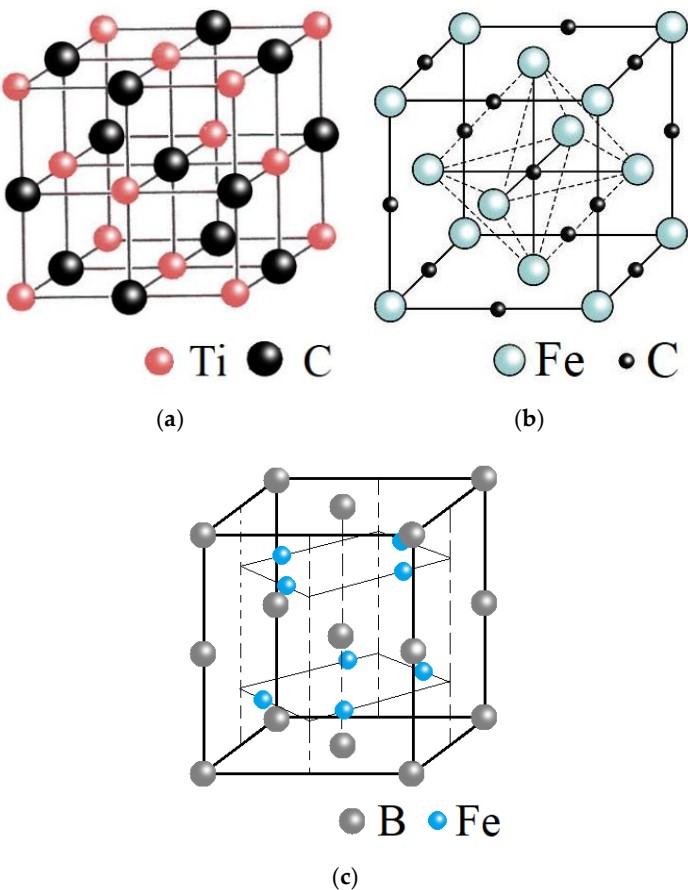

**Figure 9.** Lattice structure of TiC (**a**), austenite (**b**), and boride $Fe_2B$ (**c**).

**Table 5.** Lattice parameter of TiC, austenite, and boride $Fe_2B$ (nm).

| Phase Types | Space Group | *a* | *b* | *c* |
|:---:|:---:|:---:|:---:|:---:|
| TiC | Fm-3m | 0.433 | 0.433 | 0.433 |
| Austenite | Fm-3m | 0.357 | 0.357 | 0.357 |
| $Fe_2B$ | I4/mcm | 0.511 | 0.511 | 0.425 |

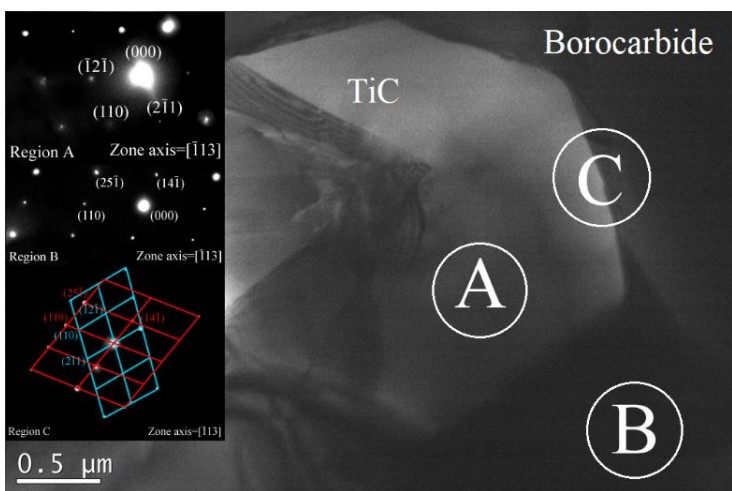

**Figure 10.** TEM bright field of TiC and $M_2(B,C)$ with corresponding selected area (electron) diffraction (SAD) patterns.

**Table 6.** Calculated results of misfit degree.

| Possible Relationship | $\delta$/% |
|---|---|
| $(110)_{TiC}//(110)_{M2(B,C)}$ | 9.3 |
| $(100)_{TiC}//(100)_{M2(B,C)}$ | 36.2 |
| $(100)_{TiC}//(110)_{M2(B,C)}$ | 23.8 |
| $(111)_{TiC}//(111)_{M2(B,C)}$ | 24.5 |
| $(111)_{TiC}//(101)_{M2(B,C)}$ | 29.1 |
| $(111)_{TiC}//(001)_{M2(B,C)}$ | 20.6 |
| $(100)_{TiC}//(100)_{\gamma}$ | 21.3 |
| $(110)_{TiC}//(110)_{\gamma}$ | 21.3 |
| $(111)_{TiC}//(111)_{\gamma}$ | 21.4 |
| $(100)_{TiC}//(110)_{\gamma}$ | 12.5 |

### 3.3. Effect of Ti Content on Mechanical Properties of High-Boron Multi-Component Alloy

Hardness is one of the most important properties of materials for abrasion. The hardness of heat-treated high-boron multi-component alloy is measured (Table 7), which indicates that after quenching and tempering, a high level of macrohardness is obtained. Moreover, Ti modification barely has an effect on macrohardness. Titanium addition results in the solid solution strengthening, thus the microhardness rises slightly.

**Table 7.** Hardness of the studied alloys.

| Hardness Types | M | T1 | T2 | T3 |
|---|---|---|---|---|
| Macrohardness after quenching/HRC | 60.6 | 64.6 | 64.9 | 62.0 |
| Macrohardness after quenching and tempering/HRC | 58.2 | 58.2 | 59.8 | 57.6 |
| Microhardness after quenching and tempering/HV | 448.9 | 468.7 | 467.6 | 484.2 |

Modification is used to improve the toughness, so it is significant to test the impact toughness of the alloys. Figure 11 shows the results of impact toughness testing of the alloys. Alloy T1, T2 and T3 possess much higher level of toughness than that of unmodified alloy M, which indicates that toughness of high-boron multi-component alloy can be remarkably increased by titanium modification. Besides, alloy T2 presents the smallest size and broken-reticular shape of borocarbide among modified alloys, but the alloy which has the highest toughness is T3. It is because T3 has the hardest matrix that it contributes to the enhance of toughness.

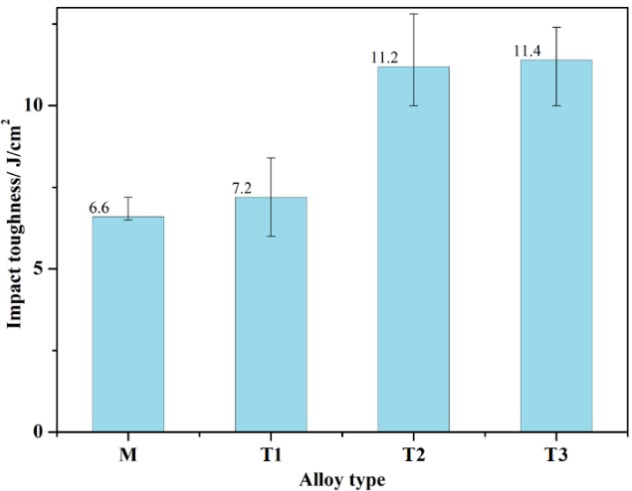

**Figure 11.** Toughness results of high-boron multi-component alloy.

The SEM SEI morphology observation of impact fractures is carried out, the results of which are shown in Figure 12. Cleavage fracture and dimples with divulsive arris are detected in each alloy. Thus it is revealed that the brittle-ductile composite fracture is the failure mechanism of high-boron multi-component alloys. In alloy M (Figure 12a), this kind of large-size fractured borocarbide is the cause of bad toughness of unmodified alloys. In Figure 12b, fractured borocarbide is seen decrescent and more dimples are observed in T1. With the further ascending of titanium content (Figure 12c,d), angles are barely observed in the fractures of alloy T2 and T3. According to all of the theoretical and experimental results, modification by titanium addition can be proved effective for improving the mechanical property of high-boron multi-component alloys with the mechanism of spheroidization and refinement of borocarbide. These results make high-boron multi-component alloys feasible for industrial application.

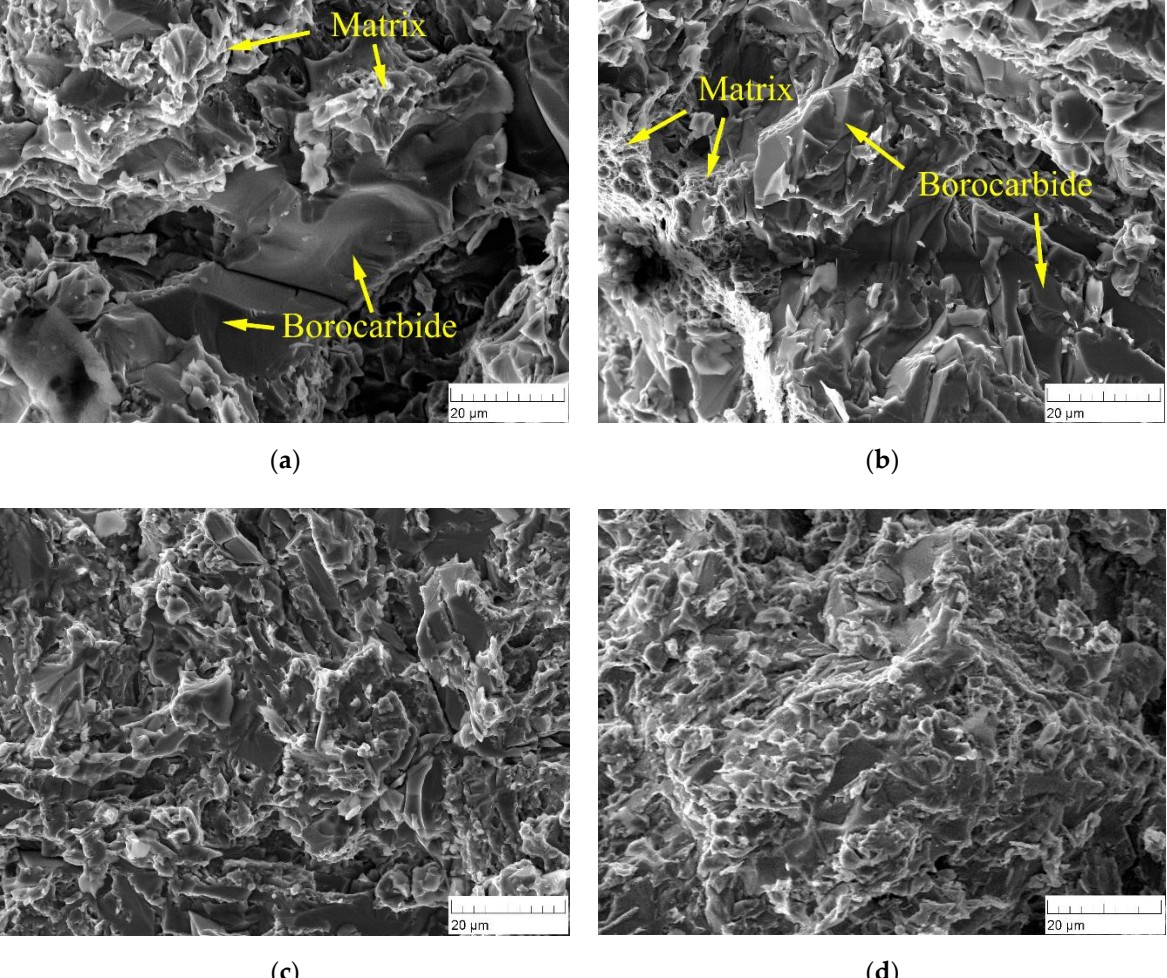

(a)       (b)

(c)       (d)

**Figure 12.** Fractures of high-boron multi-component alloy with various titanium contents: (**a**) M, 0.0 wt.%, (**b**) T1, 0.2 wt.%, (**c**) T2, 0.4 wt.%, (**d**) T3, 0.6 wt.%.

### 4. Conclusions

1. The as-cast microstructure of high-boron multi-component alloys is composed of ferrite, pearlite and $M_2(B,C)$ (M = Fe, V, Mo, Cr, Mn) (eutectic borocarbide). The un-modified borocarbide presents the structure of a continuous network. In Ti-modified high-boron multi-component alloy, the borocarbide becomes isolated, fine and spherical.

2. In liquid high-boron multi-component alloys with titanium addition, TiC particles precipitate first before solidification, which distribute dispersedly in a solidification

microstructure. With the precipitation of TiC particles, eutectic borocarbide can precipitate from where TiC particles appear. The orientation relationship of borocarbide $M_2(B,C)$ and TiC is $(110)_\gamma//(110)_{M2(B,C)}$.

3. Matrix of modified high-boron multi-component alloy is strengthened by titanium addition, the hardness of which in modified alloys is higher than that of unmodified alloys. Toughness of high-boron multi-component alloy is significantly improved by titanium modification, which keeps rising when titanium content increases. Modified alloy T3, which contains 0.6wt.% titanium, shows the best property. Fracture mechanism of high-boron multi-component alloy is mainly brittle with ductile fracture. It is the strengthening of the matrix and the improvement in the shape andsize of borocarbide that make the toughness increase.

**Author Contributions:** Conceptualization, X.R.; methodology, H.F.; software, S.T.; validation, H.F., J.X.; formal analysis, X.R.; investigation, X.R.; resources, S.T., H.F.; data curation, X.R.; writing—original draft preparation, X.R.; writing—review and editing, S.T.; visualization, H.F.; supervision, J.X.; project administration, J.X.; funding acquisition, H.F. All authors have read and agreed to the published version of the manuscript.

**Funding:** This work was supported by the National key Technologies R&D Program of China (2017ZX05009-003, 2019YFF0217501, 2019YFF0217500), Natural Science Foundation of China (U1762211) and CNPC Basic Research Project (2020B-4020, 2019B-4014, 2019E-2502).

**Conflicts of Interest:** The authors declare no conflict of interest.

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
