# Peer review of "Effect of Titanium Modification on Microstructure and Impact Toughness of High-Boron Multi-Component Alloy"

_metals, doi:10.3390/met11020193_

Round 1

Reviewer 1 Report

Some general remarks on the work “Effect of titanium modification on microstructure and toughness of high-boron multi-component alloy”.

  1. I would add “impact” before toughness in the title to avoid misunderstanding with KIC This would be consistent with the expression used by the authors in the abstract.
  2. I have some personal concerns in the use of the term “multicomponent” for this alloy, I think that if it is appropriate for the steel here investigated it would be appropriate for basically every steel. But this is just my personal opinion.
  3. Section 3.1 is unclear to me. The authors show optical micrographs and perform image analyses but I can’t really understand how images were treated. I would suggest to add one of the micrographs and its “binarized” version used for calculations.

There are some issues with verbal tenses, I am not native English speaker but there are some inconsistencies (just one example at the very beginning, row 30: “Previously, numerous metal materials with high hardness are successfully applied”).

INTRODUCTION

Row 42: I would rephrase the first sentence of the paragraph, “China is a country which is rich of boron which is worth being investigated”. This suggests that the main reason for working on this element is the availability, while the authors explained in the previous rows the importance of B addition.

Row 46: I think “reduces” would be more appropriate than “declines”

Row 51-53, please adjust the sentence “borocabide which distributes continuously severely separates the matrix [29, 32], making it unable to support the borocarbide by means of its toughness to resist the abrasion”.

Row 59: I think “natrium” is sodium and “kalium” is potassium.  

EXPERIMENTAL

Row 77: “under atmosphere”? Maybe an “inert” is missing?

Usually the authors give more details about the machinery used (make of the microscopes, and so on)

Row 96-97: the authors mention the heat treatment, but this should be said in the previous section.

The authors should clarify where the metallurgical observations were made, and at least estimate the cooling rate experienced by the materials analyzed. I think that the as cast microstructure is not homogeneous throughout the sample.

RESULTS AND DISCUSSION

Row 120: I am sorry and maybe it’s all my fault, but could the authors highlight the borocarbides in the micrographs? I don’t really understand what “reticular” means. I can clearly seen the refining effect looking at the size of ferrite islands, but it is actually hard for me to distinguish the borocarbides.

I don’t understand the effect of deep etching, if it removes the matrix why the ferrite peak is still present? Please explain which phases you are removing through deep etching.

Row 133: I don’t agree with the claim that “Formation of TiC proves that it is the necessary condition for titanium to take effect as a modifying element”. It may well be true, because there is evidence that the modifying effect is shown together with the presence of TiC, but there is no evidence that this is necessary. There could be different explanations for that.

Row 135: Maybe it’s my fault, but I can’t follow the authors. The authors comment on a martensite structure (with to me is not so clear: I expect to see martensite, but is it really that obvious in the presented micrograph, like in Fig. 5?), and on the shape of borocarbides, but the reader can’t see what is claimed.  I honestly can’t see any borocarbides, and I can’t speculate on their size or shape.

Row 150/Figure 6: I can’t understand how borocarbides volume fraction is about 30%. Could you please add an image in which you show how the different phases were discriminated?

CONCLUSIONS

In the conclusions the authors write “The as-cast  microstructure  of  high-boron  multi-component  alloy  is  composed of α-Fe (matrix) and M2(B,C)”, while in the abstract it is stated that “The results indicate that the as-cast metallurgical microstructure of high-boron multi-component alloy consists of ferrite, pearlite and borocarbide”.

Author Response

Thanks for reviewing our manuscript. Hereby is the responses for every single comment from the reviewer. Relevant correction has been clearly marked in the revised manuscript.

Responses for Reviewer:

-I would add “impact” before toughness in the title to avoid misunderstanding with KIC This would be consistent with the expression used by the authors in the abstract.

Response: Title of the revised manuscript has been changed according to the comment of reviewer.

-I have some personal concerns in the use of the term “multicomponent” for this alloy, I think that if it is appropriate for the steel here investigated it would be appropriate for basically every steel. But this is just my personal opinion.

Response: The high boron multi-component alloy is one of the ferrous alloys, but it doesn’t belong to steel or cast iron. If it is regarded as steel, it has eutectic borocarbide in its microstructure, which doesn’t confirm the definition of steel. If it is regarded as cast iron (microstructure of high boron multi-component alloy is quite similar to that of cast iron), carbon content of the alloy is lower than 2.11%, which doesn’t confirm the definition of cast iron. After consideration, we use “high boron multi-component alloy” to describe this alloy.

-Section 3.1 is unclear to me. The authors show optical micrographs and perform image analyses but I can’t really understand how images were treated. I would suggest to add one of the micrographs and its “binarized” version used for calculations.

Response: In section 2.2, Carlin corrosive that mentioned in the manuscript can change the color of matrix to black in OM image. After this treatment, different phases are clear enough to be distinguished by software ImageJ.

- There are some issues with verbal tenses, I am not native English speaker but there are some inconsistencies (just one example at the very beginning, row 30: “Previously, numerous metal materials with high hardness are successfully applied”).

Response: This may be the carelessness during typing. Relevant correction has been finished in the revised manuscript.

- Row 42: I would rephrase the first sentence of the paragraph, “China is a country which is rich of boron which is worth being investigated”. This suggests that the main reason for working on this element is the availability, while the authors explained in the previous rows the importance of B addition.

Response: To our research, two most important factors of boron is inexpensive and available. Thus the importance of boron addition is necessary to be presented.

- Row 46: I think “reduces” would be more appropriate than “declines”.

Response: The original word has been revised, which is shown in revised manuscript.

-Row 51-53, please adjust the sentence “borocabide which distributes continuously severely separates the matrix [29, 32], making it unable to support the borocarbide by means of its toughness to resist the abrasion”.

Response: Structure of this sentence has been revised, which is presented in the revised manuscript.

-Row 59: I think “natrium” is sodium and “kalium” is potassium.

Response: The original words have been changed according to the reviewer’s commment.

-Row 77: “under atmosphere”? Maybe an “inert” is missing?

Response: Inert gas wasn’t used in this melting process. The whole process is finished without gas protection.

-Usually the authors give more details about the machinery used (make of the microscopes, and so on).

Response: Other details of the equipment that used in our research are incomplete, so I’m sorry for not providing any other information.

-Row 96-97: the authors mention the heat treatment, but this should be said in the previous section.

Response: We regard heat treatment as one of the analysis methods, so introduction of it is placed in section 2.2.

-The authors should clarify where the metallurgical observations were made, and at least estimate the cooling rate experienced by the materials analyzed. I think that the as cast microstructure is not homogeneous throughout the sample.

Response: Samples for metallurgical observation are obtained from the center of the ingot, which has been revised in row 82. Cooling rate is presented in row 98 in the revised manuscript. Ingot that we used in this work is small enough, inhomogeneous distribution of microstructure can’t happen during its solidification.

-Row 120: I am sorry and maybe it’s all my fault, but could the authors highlight the borocarbides in the micrographs? I don’t really understand what “reticular” means. I can clearly seen the refining effect looking at the size of ferrite islands, but it is actually hard for me to distinguish the borocarbides.

Response: In as-cast microstructure images, ferrite and borocarbide are all white, for showing the morphology of matrix, observation of borocarbide is influenced. So it is hard to distinguish the reticular structure of borocarbide. In heat-treated microstructure images, matrix becomes dark, thus the continuous network of borocarbide (Fig. 5a) is clear enough to be seen.

-I don’t understand the effect of deep etching, if it removes the matrix why the ferrite peak is still present? Please explain which phases you are removing through deep etching.

Response: Deep etching can remove about 0.5mm of the matrix (include all of the phases except borocarbide) from the surface that prepared for XRD analysis. This treatment can effectively weak the strength of matrix in XRD pattern rather than completely remove it. So peak of matrix can still be detected in the pattern of deep etched sample.

-Row 133: I don’t agree with the claim that “Formation of TiC proves that it is the necessary condition for titanium to take effect as a modifying element”. It may well be true, because there is evidence that the modifying effect is shown together with the presence of TiC, but there is no evidence that this is necessary. There could be different explanations for that.

Response: The original description has been changed to words which are more accurate in the revised manuscript.

-Row 135: Maybe it’s my fault, but I can’t follow the authors. The authors comment on a martensite structure (with to me is not so clear: I expect to see martensite, but is it really that obvious in the presented micrograph, like in Fig. 5?), and on the shape of borocarbides, but the reader can’t see what is claimed.  I honestly can’t see any borocarbides, and I can’t speculate on their size or shape.

Response: For the clear observation of borocarbide, the magnification is too low to see the structure of martensite clearly. In Fig. 5, the white area is borocarbide.

-Row 150/Figure 6: I can’t understand how borocarbides volume fraction is about 30%. Could you please add an image in which you show how the different phases were discriminated?

Response: During the measuring process of volume fraction, contrast of the tested image is ascended, matrix changes to black and borocarbide keeps white. It is clear enough for software to distinguish the white area and calculate the area fraction, like the images as follows(Figures are in the uploaded file).

-In the conclusions the authors write “The as-cast  microstructure  of  high-boron  multi-component  alloy  is  composed of α-Fe (matrix) and M2(B,C)”, while in the abstract it is stated that “The results indicate that the as-cast metallurgical microstructure of high-boron multi-component alloy consists of ferrite, pearlite and borocarbide”.

Response: It’s my fault that giving the different conclusions. Relevant correction has been added in the revised manuscript.

Reviewer 2 Report

The submitted manuscript " Effect of titanium modification on microstructure and toughness of high-boron multi-component alloy "deals with interesting subject of process-microstructure-property relations in multi-component cast steel. The submitted manuscript is easy to read and, in general, is well organised. The scientific quality of the paper is reasonable, and the subject is of significance for readers. On the other hand, despite high importance of the investigated problems, the paper is rather validating achievements in the subject. On the other hand, the readability of the paper in current form is quite good, and the scientific quality considerable. Some fragments need polishing as some sentences are not clear and unambiguous, and are suggested for improvement for better readability.

As indicated by the authors, the subject of mechanical properties improvement, such as impact strength, is a vital problem,  and there are many aspects to attract the interest. Thus I find the subject interesting for both scientists and industry.   However, there are issues to amend and manuscript needs significant corrections to improve readability and comprehensibility. The text should be revised; as there are many style imperfections, which  deteriorate the value of presented findings.

The remarks listed below are to help in this respect:

1) A relatively high number of auto-citing references is included, which in  my opinion is not justified by importance of statements concerned (e.g. mentioning what authors were investigating in previous papers - not really related to title). The auto-references are advised to be limited to those related with the subject directly.

Introduction to the study is comprehensive and sufficient for the problem, however, again, the proportions between references are askew - there a many references concerning the material (multi-component boron cast steel), with only a few concerning effect of titanium modification.

2) Discussion of the results is chaotic and it is not quite clear, if some conclusions concern the authors results or are citation of general knowledge. For instance, lines 21-26 ("Some of the previous works 121 [36, 40, 42, 44, 45] have indicated that titanium in liquid steel forms plenty of intermetallic  compound particles with other elements. These particles act as heterogeneous nuclei, providing more base for nucleation of borocarbide and significantly rise its nucleation 124 rate. When amount of borocarbide is constant, the higher nucleation rate borocarbide has, the more isolated borocarbide will appear..."), or lines 153-159,  sounds like statements of literature based knowledge.  Essentially, the explanation is correct, but the manner in which it's written should be concise and more directly related to results.

It’s advised to move the knowledge to introduction or address the literature to results in a more consistent way.

3) there are numerous errors in the text, including:

i) grammar and style errors in sentences:

  • (line) 208: Data in Table 5 is details of them
  • 213: the results that shown in Table 6.
  • 199: are the symbol of
  • 202, 203: which from (110)TiC//(110)M2
  • caption of Fig. 6
  • 85: analyses

ii) there are fragments which despite being correct, sound unambiguous, no good or have some style imperfections, or need clarification e.g.:

  • line 42: "which is rich of boron which is worth being investigated"
  • line 265: "hardness of which in modified alloy is higher than that of unmodified alloy. Toughness of high-boron multi-component  alloy is significantly improved by titanium modification, which keeps rising when titanium content increases" - revise to remove repeatition
  • 77: under atmosphere (detail)
  • 96: deeply etched
  • 222: This work measured the hardness - try passive voice, without personification of work
  • 232: contributes to the  enhance of toughness
  • 239: brittle-ductile composite fracture is the mechanism of high-boron multi-component alloy. - mechanism of what?
  • 212: and the centric spot are detected alined obviously  -revise
  • 246: make high-boron multi-component alloy for industrial application feasible - 1) structure of the sentence, 2) in addition to feasibility it is worth mentioning whether it is economically beneficial
  • 258: TiC particles precipitate first before solidification - are first to precipitate (?)
  • 260: distribute dispersedly in solidification microstructure  - (microstructure after solidification ?)
  • -263: is strengthened because of titanium addition - because of or by ?
  • 268: performs the best property - perform the best or shows/exhibits best properties

Author Response

Thanks for reviewing our manuscript. Hereby is the responses for every single comment from the reviewer. Relevant correction has been clearly marked in the revised manuscript.

Responses for Reviewer:

- Introduction to the study is comprehensive and sufficient for the problem, however, again, the proportions between references are askew - there a many references concerning the material (multi-component boron cast steel), with only a few concerning effect of titanium modification.

Response: Extra references about titanium modification and other elements’ modification has been added in the revised manuscript.

- It’s advised to move the knowledge to introduction or address the literature to results in a more consistent way.

Response: Details of the introduction has been revised according to the reviewer’s comment.

- grammar and style errors in sentences:

(line) 208: Data in Table 5 is details of them

213: the results that shown in Table 6.

199: are the symbol of

202, 203: which from (110)TiC//(110)M2

caption of Fig. 6

85: analyses.

Response: Grammar and spelling mistakes has been corrected in the revised manuscript.

- there are fragments which despite being correct, sound unambiguous, no good or have some style imperfections, or need clarification e.g.:

line 42: "which is rich of boron which is worth being investigated"

line 265: "hardness of which in modified alloy is higher than that of unmodified alloy. Toughness of high-boron multi-component  alloy is significantly improved by titanium modification, which keeps rising when titanium content increases" - revise to remove repeatition

Response: Relevant correction has been presented in the revised manuscript.

77: under atmosphere (detail)

96: deeply etched

222: This work measured the hardness - try passive voice, without personification of work

Response: The original sentence has been corrected to passive voice.

232: contributes to the  enhance of toughness

239: brittle-ductile composite fracture is the mechanism of high-boron multi-component alloy. - mechanism of what?

Response: Sorry for missing the word. Correction has been finished in the revised manuscript.

212: and the centric spot are detected alined obviously  -revise

246: make high-boron multi-component alloy for industrial application feasible - 1) structure of the sentence, 2) in addition to feasibility it is worth mentioning whether it is economically beneficial

Response: Information about whether it is economically beneficial has been presented in the details of introduction.

258: TiC particles precipitate first before solidification - are first to precipitate (?)

Response:

260: distribute dispersedly in solidification microstructure  - (microstructure after solidification ?)

Response: Yes, solidification microstructure means the as-cast microstructure after solidification.

263: is strengthened because of titanium addition - because of or by ?

Response: Sorry for not describing it accurately. Ti addition causes the solid solution strengthening of matrix. So matrix is strengthened by Ti addition. Relevant correction has been finished in the revised manuscript.

268: performs the best property - perform the best or shows/exhibits best properties.

Response: The original word has been changed in the revised manuscript.

Round 2

Reviewer 1 Report

(in the uploaded file I used colors for a better reading. I deleted all the comments which were addressed in the revised version)

-Row 51-53, please adjust the sentence “borocabide which distributes continuously severely separates the matrix [29, 32], making it unable to support the borocarbide by means of its toughness to resist the abrasion”.

Response: Structure of this sentence has been revised, which is presented in the revised manuscript.

There is a typo in row 50 (“Howerver”)

-Row 77: “under atmosphere”? Maybe an “inert” is missing?

Response: Inert gas wasn’t used in this melting process. The whole process is finished without gas protection.

Please consider the use of “in air”. When I read “atmosphere” I immediately thought that something was missing. This is only a suggestion, just think about it.

-Usually the authors give more details about the machinery used (make of the microscopes, and so on).

Response: Other details of the equipment that used in our research are incomplete, so I’m sorry for not providing any other information.

It is not really a big deal to me, but the policy of the journal usually requires more details about the machinery, so maybe the editor will ask you to provide them. Anyway, to me it is not a problem.

-Row 96-97: the authors mention the heat treatment, but this should be said in the previous section.

Response: We regard heat treatment as one of the analysis methods, so introduction of it is placed in section 2.2.

I only notice it now, but at row 91 you write “selecting area diffraction”, which should read “selected area (electron) diffraction”.

Heat treatment is not an analysis method. You tested the impact toughness of heat treated material, so the results of the impact test are “analysis”, but what is the result of the heat treatment? The result is the material, which you then analyzed. Moreover, you analyzed both untreated and treated material, so you should give this information in section 2.1. I know you basically wrote the same description in another paper (“Effect of calcium modification on solidification, heat treatment microstructure and toughness of high boron high speed steel”, in “Materials Research Express”), but I think it was not the best choice in that case either.

-The authors should clarify where the metallurgical observations were made, and at least estimate the cooling rate experienced by the materials analyzed. I think that the as cast microstructure is not homogeneous throughout the sample.

Response: Samples for metallurgical observation are obtained from the center of the ingot, which has been revised in row 82. Cooling rate is presented in row 98 in the revised manuscript. Ingot that we used in this work is small enough, inhomogeneous distribution of microstructure can’t happen during its solidification.

I would suggest to place here the estimated cooling rate instead of section 2.2 (something like “All of the specimens were cut from the center of the ingot, where the estimated cooling rate is 40°C/s). The importance of this information is lower for the treated material, which I believe is more homogeneous.

-Row 120: I am sorry and maybe it’s all my fault, but could the authors highlight the borocarbides in the micrographs? I don’t really understand what “reticular” means. I can clearly seen the refining effect looking at the size of ferrite islands, but it is actually hard for me to distinguish the borocarbides.

Response: In as-cast microstructure images, ferrite and borocarbide are all white, for showing the morphology of matrix, observation of borocarbide is influenced. So it is hard to distinguish the reticular structure of borocarbide. In heat-treated microstructure images, matrix becomes dark, thus the continuous network of borocarbide (Fig. 5a) is clear enough to be seen.

I am afraid that this is still completely unclear to me. Looking at Fig. 5, ferrite is white while basically everything else is borocarbide phase? Could you highlight “one single borocarbide”? To me, the non-white phases appears connected, so I can’t honestly understand how you can “isolate” borocarbide phases to measure their size. My request of a “binarized image” aimed at showing this information, could you show one of them to me, if you think it would make the paper too long?

-Row 135: Maybe it’s my fault, but I can’t follow the authors. The authors comment on a martensite structure (with to me is not so clear: I expect to see martensite, but is it really that obvious in the presented micrograph, like in Fig. 5?), and on the shape of borocarbides, but the reader can’t see what is claimed.  I honestly can’t see any borocarbides, and I can’t speculate on their size or shape.

Response: For the clear observation of borocarbide, the magnification is too low to see the structure of martensite clearly. In Fig. 5, the white area is borocarbide.

-Row 150/Figure 6: I can’t understand how borocarbides volume fraction is about 30%. Could you please add an image in which you show how the different phases were discriminated?

Response: During the measuring process of volume fraction, contrast of the tested image is ascended, matrix changes to black and borocarbide keeps white. It is clear enough for software to distinguish the white area and calculate the area fraction, like the images as follows(Figures are in the uploaded file).

At Row 119 I read “It is observed that matrix in alloy consists of ferrite (light area) and pearlite (dark area)”, referring to Fig. 3. Instead, if I got it right, in fig. 5 the grey areas are martensite and the white ones are borocarbides. Is it correct? If so, it is indeed easy to distinguish these two phases in the treated material.

However, you say that the as-cast microstructure shows borocarbide, pearlite and ferrite, and I still can’t understand how you determined the volume fraction of borocarbides in this condition. If it is an effect of Carlin corrosive, I am afraid but this should be explained better.

I am sorry but I don’t have access to any uploaded file and I can’t see anything but what is in the paper.

Author Response

Thanks for reviewing our manuscript. Hereby is the responses for every single comment from the reviewer. Relevant correction has been clearly marked in the revised manuscript.

Responses for Reviewer:

-There is a typo in row 50 (“Howerver”).

Response: It’s my mistake. Relevant correction has been finished in the revised manuscript.

-Please consider the use of “in air”. When I read “atmosphere” I immediately thought that something was missing. This is only a suggestion, just think about it.

Response: Comment that indicated by reviewer is more proper. The original word has been changed.

-It is not really a big deal to me, but the policy of the journal usually requires more details about the machinery, so maybe the editor will ask you to provide them. Anyway, to me it is not a problem.

Response: We have done our best on this problem. Some of the machineries have been obtained and presented in the revised manuscript.

-I only notice it now, but at row 91 you write “selecting area diffraction”, which should read “selected area (electron) diffraction”.

Response: Words have been revised as reviewer’s comments.

Heat treatment is not an analysis method. You tested the impact toughness of heat treated material, so the results of the impact test are “analysis”, but what is the result of the heat treatment? The result is the material, which you then analyzed. Moreover, you analyzed both untreated and treated material, so you should give this information in section 2.1. I know you basically wrote the same description in another paper (“Effect of calcium modification on solidification, heat treatment microstructure and toughness of high boron high speed steel”, in “Materials Research Express”), but I think it was not the best choice in that case either.

Response: Description about heat treatment has been moved to section 2.1, which is more logically suitable.

-I would suggest to place here the estimated cooling rate instead of section 2.2 (something like “All of the specimens were cut from the center of the ingot, where the estimated cooling rate is 40°C/s). The importance of this information is lower for the treated material, which I believe is more homogeneous.

Response: The relevant sentences have been revised as reviewer’s comments.

-I am afraid that this is still completely unclear to me. Looking at Fig. 5, ferrite is white while basically everything else is borocarbide phase? Could you highlight “one single borocarbide”? To me, the non-white phases appears connected, so I can’t honestly understand how you can “isolate” borocarbide phases to measure their size. My request of a “binarized image” aimed at showing this information, could you show one of them to me, if you think it would make the paper too long?

Response: Image in the uploaded PDF file is the as-cast microstructure which are dyed by Carlin corrosive. Black area is matrix, light area is borocarbide. It is clear enough for software to distinguish.

-At Row 119 I read “It is observed that matrix in alloy consists of ferrite (light area) and pearlite (dark area)”, referring to Fig. 3. Instead, if I got it right, in fig. 5 the grey areas are martensite and the white ones are borocarbides. Is it correct? If so, it is indeed easy to distinguish these two phases in the treated material.

However, you say that the as-cast microstructure shows borocarbide, pearlite and ferrite, and I still can’t understand how you determined the volume fraction of borocarbides in this condition. If it is an effect of Carlin corrosive, I am afraid but this should be explained better.

I am sorry but I don’t have access to any uploaded file and I can’t see anything but what is in the paper.

Response: We have presented the binarized image in the previous response letter, unfortunately, the uploaded file is missing. Images that reviewer want are in the uploaded PDF file.
